# Biosynthesis of Quercetin-Loaded Melanin Nanoparticles for Improved Antioxidant Activity, Photothermal Antimicrobial, and NIR/pH Dual-Responsive Drug Release

**DOI:** 10.3390/foods12234232

**Published:** 2023-11-23

**Authors:** Di Zhang, Xianrui Chen, Nitong Bu, Liying Huang, Huanglong Lin, Lizhen Zhou, Ruojun Mu, Lin Wang, Jie Pang

**Affiliations:** 1College of Food Science, Fujian Agriculture and Forestry University, Fuzhou 350002, Chinamu@fafu.edu.cn (R.M.); 2Department of Engineering Mechanics, Tsinghua University, Beijing 100084, China

**Keywords:** melanin nanoparticles, quercetin, antioxidant activity, photothermal antimicrobial, stimuli drug release

## Abstract

Quercetin (QCT) is a promising dose-dependent nutraceutical that usually suffers from poor water solubility and low bioavailability issues. In this work, a novel QCT-loaded nanoscale delivery system was constructed based on the oxidative self-polymerization of melanin (Q@MNPs). The FT-IR, XRD, and Zeta potential analyses confirmed that QCT was successfully absorbed on the melanin nanoparticles (MNPs) via Π−Π and hydrogen bonding interactions. The encapsulation efficiency and particle size of Q@MNPs were 43.78% and 26.68 nm, respectively. Q@MNPs improved the thermal stability of QCT and the antioxidant properties in comparison to MNPs. Meanwhile, Q@MNPs presented fantastic photothermal conversion capacity and stability triggered by the NIR laser, which significantly enhanced the antibacterial capability with a sterilization rate of more than 98% against *E. coli* and *S. aureus*. More importantly, Q@MNPs exhibited NIR/pH dual-responsive drug release behavior and good biocompatibility (at concentrations of < 100 μg/mL). Thus, Q@MNPs show promising prospects for flavonoid delivery.

## 1. Introduction

Food spoilage and deterioration caused by bacterial infection are still a challenge in the world. There are considerable concerns about developing and designing antimicrobial materials for food preservation. Finding safe antibacterial materials to reduce environmental pollution is a current trend. Quercetin (QCT) is an abundant natural flavonoid found in many common fruits and vegetables with many bioactivities, including antioxidant, antibacterial, and anticancer activities, and it is used in the food and medical fields. Arcan et al. incorporated QCT into zein films as bioactive packaging materials [1]. George et al. found that QCT has anticancer activities in cancerous cells [2]. However, the poor water solubility and instability of QCT during processing limits its application. With the development of nanotechnology, many advanced research paths have opened up in different fields. Nanotechnology is a constantly evolving and innovative technology that can be used to develop nanoparticles (NPs), which are widely used for encapsulating functional compounds, such as essential oils, antimicrobial agents, and natural extracts [3,4]. Moreover, it can improve the stability of functional compounds, control delivery, and enhance bioactivity. Nanoparticles have been used for the delivery of QCT. To improve its bioavailability, organic solvents like dimethyl sulfoxide (DMSO) have been used to produce nanoparticles loaded with QCT [5], whereas the safety of organic solvents is a potential issue. Some methods use chemical crosslinking to prepare nanoparticles loaded with QCT [6]. However, the crosslinking agents used to prepare chemically crosslinked nanoparticles have toxicity concerns. Hence, natural polymer-based nanomaterials, including proteins and polysaccharides, were further employed to fabricate nanoparticles to encapsulate QCT. For example, zein nanoparticles have been reported to encapsulate QCT, although the stability of zein nanoparticles is far from satisfactory. Therefore, researchers attempted to add polysaccharides to enhance the stability of the nanoparticles. Zou et al. encapsulated QCT in chitosan-coated zein nanoparticles to enhance its water dispersibility, antioxidant activity, and bio-accessibility [7]. However, it is still a challenge to create nanoparticles with controlled and stimuli-responsive abilities to release QCT.

Among these nanoparticles, melanin-like nanoparticles (MNPs) have received much interest because of their biocompatibility, biodegradability, free radical scavenging ability, and photothermal properties with strong absorption of near-infrared (NIR) laser and conversion of light energy to heat energy [8]. Therefore, it is widely used in food, energy, and medical fields. MNPs can be produced from natural microorganisms, animals, and plants [9]. However, the practical applications of natural melanin are limited due to its poor purity and extraction rate [10]. Another alternative method is the use of a green synthetic for dopamine oxidation. The dopamine hydrochloride can dissolve and oxidize in an alkaline aqueous solution to form MNPs, which demonstrate comparable physical and chemical properties compared to those of natural MNPs [11]. It has been proven that MNPs can be directly modified using aromatic molecules and further applied as surface modification agents for nanoparticles, filler materials for light absorption, and drug-controllable and -stimulating delivery platforms [12]. Specifically, there are many conjugated polymeric structures and reactive functional groups, including quinone groups, phenolic hydroxyl, hydroxyl groups, and amino groups in MNPs, that enable it to easily modify the surface and combine with the aromatic structures of substances via Π−Π stacking and hydrogen bonding interactions [13]. In previous studies, Xing et al. formed core-shell lignin-melanin NPs (LMNP) using in situ polymerization of dopamine on the surface of lignin nanoparticles [14]. Liu et al. synthesized silver/melanin-like nanoparticles (Ag-SMNPs) to fabricate cotton fabric with multifunctional UV-resistant, antibacterial, and photothermal properties [15]. Recently, Wang et al. extracted natural melanin to produce MNPs and encapsulated QCT on the surface of MNPs, which exhibited excellent antibacterial and anti-tumor properties [16], improved the bioavailability of QCT, and expanded the application of natural MNPs. However, related research on the photothermal properties and drug release behavior of QCT-loaded MNPs from the green synthesis of dopamine oxidation is lacking.

In this work, a novel functional nano-carrier of dopamine hydrochloride-derived MNPs was fabricated and loaded with QCT to obtain Q@MNPs (Figure 1a). Fourier-transform infrared spectroscopy (FT-IR), X-ray diffraction analysis (XRD), thermo-gravimetric analysis (TGA), transmission electron microscopy (TEM), and Zeta potential values were used to characterize Q@MNPs. The photothermal performance and stability of Q@MNPs were evaluated using photothermal experiments. The antioxidant activities of Q@MNPs, and antibacterial activities of Q@MNPs combining photothermal effects against *S. aureus* and *E. coli* were explored. The pH and NIR dual stimuli release of QCT and the biocompatibility of Q@MNPs were also investigated. The findings of this research can broaden the application of melanin nanoparticles in food processing and provide an appealing method to produce specific drug delivery systems.

## 2. Materials and Methods

### 2.1. Materials

Dopamine hydrochloride (>98%, purity), 2,2-Diphenyl-1-picrylhydrazyl (DPPH), 2,2′-azino-bis (3-ethylbenzothiazoline-6-sulfonic acid) (ABTS), and potassium persulfate (K_2_S_2_O_8_) were obtained from Aladdin Biochemical Technology Co. Ltd. (Shanghai, China). Quercetin (97%, purity) was purchased from Maclin Biochemical Technology Co., Ltd. (Shanghai, China). *Escherichia coli* (*E. coli*) and *Staphylococcus aureus* (*S. aureus*) were supplied by the food microbiology laboratory at the College of Food Science, Fujian Agriculture and Forestry University (Fuzhou, China). The ultrapure water was used in this experiment, and all other reagents used were of analytical grade.

### 2.2. Preparation of MNPs

The MNPs were produced based on a previous method, with slight modifications [12]. In brief, 2 g of dopamine hydrochloride was dissolved in 200 mL of deionized water under magnetic stirring at 600 rpm at 40 °C. After dopamine hydrochloride was completely dissolved, 6 mL of 1 mol/L NaOH was added to the solution. During the reaction, the color of the solution changed from light yellow to brownish-black. The solution was continuously stirred at this temperature in the dark for 24 h. After polymerization, the synthesized MNPs were obtained via centrifugation at 11,000× *g* and washed with deionized water. This procedure was repeated several times until the supernatant was pale and neutral. Subsequently, the precipitate was dried in an oven (40 °C) for 24 h to obtain MNP powder.

### 2.3. Preparation of Q@MNPs

First, QCT (200 mg) was weighed and dissolved in 100 mL of anhydrous ethanol. Then, the prepared MNP powder (100 mg) was added to the solution with magnetic stirring (600 rpm) in the dark at 25 °C for 12 h. After stirring, the Q@MNPs were obtained via centrifugation at 11,000× *g*. The precipitate was dried in an oven (40 °C) for 12 h to obtain Q@MNPs powder. Eventually, the Q@MNPs powder was stored at 4 °C until further experiments.

#### Encapsulation Efficiency (EE) of QCT

The above-mentioned supernatant of Q@MNPs was measured to determine the absorbance of free QCT spectrophotometrically at 373 nm using a UV-vis spectrophotometer (UV-2600, Shimadzu, Japan). The standard curve of QCT dissolved in anhydrous ethanol was measured ($y=0.0707\;x+0.0365,\;R^{2}=0.9992$). The amount of free QCT was evaluated using a standard curve. The encapsulation efficiency (EE) of QCT was calculated as follows:$$EE\;(\%)=\frac{m_{t}-m_{f}}{m_{t}}\times 100$$
where $m_{t}$ was the total amount of QCT and $m_{f}$ was the free amount of QCT.

### 2.4. Characterization of Q@MNPs

#### 2.4.1. Fourier-transform Infrared Spectroscopy (FT-IR)

The FT-IR spectra of QCT, MNPs, and Q@MNPs were measured using a Bruker Vertex 70 FT-IR spectrometer (Karlsruhe, Germany) using the KBr tableting method, and the scanning range was from 4000 to 400 cm^−1^ with a resolution of 4 cm^−1^.

#### 2.4.2. X-ray Diffraction Analysis (XRD)

The XRD spectra of QCT, MNPs, and Q@MNPs were obtained using an X-ray diffractometer (D8, Bruker, Germany) with a diffraction angle of 10–80° and at a scanning speed of 5°/min.

#### 2.4.3. Thermo-Gravimetric Analysis (TGA)

The thermal stability of QCT, MNPs, and Q@MNPs was investigated using a thermal analyzer (Q500 TGA, Newcastle, DE, USA) heated at a rate of 10 °C/min with a scan range from 30 to 600 °C under a nitrogen atmosphere.

#### 2.4.4. Transmission Electron Microscopy (TEM)

The morphology of Q@MNPs was characterized using TEM (JEOL JEM 2100F, Japan Electronics Co., Ltd., Tokyo, Japan).

#### 2.4.5. Zeta Potential

Based on the method of Wang et al. [17], 10 mg of MNPs and QCT@MNPs powder was dispersed in 10 mL of deionized water and ultrasonicated for 10 min. Then, the Zeta potential of the MNPs and Q@MNPs solutions was examined using a Litesizer 500 (Anton Parr, Graz, Austria) at 25 °C, and three parallel measurements were performed.

#### 2.4.6. Particle Size of Q@MNPs

A Litesizer 500 (Anton Parr, Graz, Austria) was used to measure the particle size and polydispersity index (PDI) of Q@MNPs at 25 °C. The sample was dispersed and diluted with deionized water before observation.

### 2.5. Photothermal Performance

The photothermal performance of Q@MNPs was tested using the method described by Liang et al., with some modifications [8]. In brief, the Q@MNPs powder was weighed (0.1 g) and pressed into a disc with a diameter of 1.3 cm. Then, the sample was irradiated by an 808 nm laser (0.25 W/cm^2^) for 10 min. At the same time, the thermograph and the infrared camera were used to record the temperature on the surface of the samples. To evaluate the photothermal stability of Q@MNPs, the sample was exposed to an 808 nm laser, irradiated, and cooled for 10 min. The heating–cooling curves were obtained for Q@MNPs over 4 cycles.

### 2.6. Antioxidant Activity of Q@MNPs

#### 2.6.1. DPPH Radical Scavenging Activity

Following the method described by Zhao et al. [18], the DPPH radical scavenging activity of Q@MNPs was measured. In brief, different quantities of Q@MNPs were weighed (0.5, 1, 2, 4, and 8 mg) and then immersed in 10 mL DPPH anhydrous ethanol solution to ensure that the concentrations of Q@MNPs solutions were 0.05, 0.1, 0.2, 0.4, and 0.8 mg/mL (*w*/*v*), respectively. Then, the solution was allowed to react in the dark at room temperature for 15 min. After the reaction, the absorbance of the solutions was measured at 517 nm using a UV-vis spectrophotometer (UV–2600, Shimadzu, Japan). The DPPH radical scavenging ability of different concentrations of Q@MNPs was determined using the following equation:$$Scavenging\;ability\;of\;DPPH\;(\%)=\frac{A_{0}-A_{i}}{A_{0}}\times 100$$
where $\;A_{0}$ and $A_{i}$ are the absorbance of the blank and samples, respectively.

#### 2.6.2. ABTS Scavenging Activity

The ABTS radical scavenging rate of Q@MNPs was measured according to a method described in a previous study, with slight modifications [19]. Briefly, an equal volume of 7 mM ABTS and 4.9 mM potassium persulfate was mixed and stored at 25 °C in the dark for 16 h. Subsequently, the ABTS/potassium persulfate mixture was diluted with anhydrous ethanol to obtain the absorbance of ABTS solution, 0.7 ± 0.02, at 734 nm. After that, different quantities of Q@MNPs were weighed and then immersed in 10 mL ABTS solution to obtain different concentrations of Q@MNPs solutions (0.05, 0.1, 0.2, 0.4, and 0.8 mg/mL). The solution was allowed to react in the dark at 25 °C for 5 min. After the reaction, the absorbance of the solutions was determined at 734 nm using a UV-vis spectrophotometer (UV-2600, Shimadzu, Japan). The ABTS radical scavenging activity of different concentrations of Q@MNPs was determined as follows:$$Scavenging\;ability\;of\;ABTS\;(\%)=\frac{A_{0}^{1}-A_{i}^{1}}{A_{0}^{1}}\times 100$$
where $\;A_{0}^{1}$ is the absorbance of the blank and $A_{i}^{1}$ is the absorbance of the samples.

### 2.7. Antibacterial Activity

*E. coli* and *S. aureus* were applied to test the antibacterial activity of Q@MNPs using a previously reported method, with slight modifications [20]. The quantitative of QCT, MNPs, and Q@MNPs were ultrasonically dispersed in a phosphate-buffered saline (PBS) at pH 7.2 to make QCT, MNPs, and Q@MNPs solutions at a concentration of 200 μg/mL (*m*/*v*). This test was divided into four groups for comparison: control (PBS), QCT, MNPs, and Q@MNPs. Each group was divided into two treatment groups, including near–infrared light irradiation and non–irradiation. Subsequently, 100 μL of the bacterial suspension (1 × 10^6^ CFU/mL) was added to a 24-well plate. Subsequently, 900 μL of PBS, QCT, MNPs, and Q@MNPs solutions were mixed with the bacterial suspension, and the mixture was incubated at 37 °C for 1 h. For each group that needed contact, near–infrared light irradiation was applied using an 808 nm laser with 1.0 W/cm^2^ for 10 min. Then, 100 μL of the treated bacterial suspension was taken out and inoculated on LB agar plates at 37 °C for 16 h. Eventually, the number of bacteria was counted to show the antibacterial performance of Q@MNPs. The experiments of each sample were performed in triplicate. The number of bacterial colonies in each sample was estimated using the following equation:$$Bacterial\;survival\;(\%)=\frac{N_{i}}{N_{0}}\times 100$$
where $N_{i}$ is the number of bacterial colonies in the QCT, MNPs, and Q@MNPs groups without or with NIR, respectively; $N_{0}$ is the number of bacterial colonies in the control group.

### 2.8. In Vitro QCT Release

The QCT release experiment was conducted using a previously described method, with some modifications [21], and the Q@MNPs were divided into two groups at different pH (4.5 and 7.2). Each group was divided into two treatment groups, including near–infrared light (808 nm, 1.0 W/cm^2^) irradiation for 10 min and non–irradiation before the first extraction. The Q@MNPs (30 mg) were weighed and dispersed in 40 mL of PBS solution. Subsequently, it was placed in a constant-temperature shaker at 80 rpm and 37 °C. At the predetermined time, 4 mL of the mixed solution was collected, and then an equivalent volume of fresh PBS solution was replaced. The extract solution was centrifuged at 5000× *g* for 5 min, and the absorbance of the supernatant was measured at 373 nm using a UV-vis spectrophotometer (UV-2600, Shimadzu, Japan). The cumulative release rate of QCT was calculated as follows:$$Cumulative\;release\;(\%)=\frac{V_{0}\times C_{n}+V_{1}\times \sum C_{n-1}}{m_{0}}\times 100$$
where $m_{0}$ is the weight of QCT in the sample, and $V_{0}$ and $V_{1}$ are the total volume and extraction volume of the PBS solutions at the predetermined time, respectively. The $C_{n}$ and $C_{n-1}$ represent the concentrations of QCT in the n and n−1 releases of PBS solutions, respectively.

### 2.9. In Vitro Cytotoxicity Assay

According to the reference method with some modifications [22], the relative cell cytotoxicity of Q@MNPs was evaluated via CCK–8 assay using L929 cells. Quantitative nanoparticles were sterilized using ultraviolet irradiation for 30 min and ultrasonically dispersed in MEM medium solution for 5 min to prepare different concentrations of Q@MNPs solutions. The 100 μL of fresh MEM medium solution containing different concentrations of Q@MNPs (50 μg/mL, 100 μg/mL, 200 μg/mL, and 500 μg/mL) was added to 96-well plates, and 100 μL of fresh MEM culture medium solution was added to the 96-well plates as the control group. Then, L929 cells were added to the plates at a density of 8 × 10^3^ cells/well and cultured for 24 h (5% CO_2_, 37 °C). Subsequently, the culture medium was removed and cleaned with PBS solution three times, and 150 μL of MEM medium solution containing 10% CCK−8 was added and incubated in a constant-temperature incubator (5% CO_2_, 37 °C). After 2 h, 100 μL of the supernatant from each well was transferred to a new plate. Finally, the absorbance at 450 nm was recorded using a microtiter reader (SPARK 10M, TECAN, Männedorf, Austria). Each experiment was performed in triplicate.

### 2.10. Statistical Analysis

All results are expressed as the mean ± standard deviation (SD). GraphPad Prism 9 (GraphPad Software, San Diego, CA, USA) was used to analyze the significance of the differences (*p* < 0.05) using the one-way analysis of variance (ANOVA).

## 3. Results and Discussion

### 3.1. FT-IR Analysis

FT-IR spectrometry was used to study the structural changes in MNPs after binding to QCT molecules, as presented in Figure 1b. For the characteristic bands of QCT, 3409 cm^−1^ was assigned to O–H stretching, and the band situated at 1664 cm^−1^ was assigned to C=O stretching vibration. The characteristic bands that appeared at 1611 cm^−1^ and 1521 cm^−1^ were the stretching vibrations of the benzene ring skeleton [23]. And the characteristic band at 1014 cm^−1^ corresponded to C–O and C–C–O stretching vibrations [24]. In the spectra of MNPs, the bands at 1619 cm^−1^ and 1513 cm^−1^ were linked to the C=C bond of the benzene rings and C–N in indolequinone, respectively [25]. The band at 1247 cm^−1^ was linked to the C–O–H bending and stretching of phenolics [26]. Moreover, when QCT was combined with MNPs, the peaks of the Q@MNPs changed. The peak at 3416 cm^−1^ shifted to 3404 cm^−1^, and the intensity of the peaks at 1617 and 1512 cm^−1^ increased. Furthermore, the Q@MNPs showed a new peak at 1013 cm^−1^, which was the characteristic band of QCT. This can be ascribed to the hydrogen bonding and Π−Π stacking of the aromatic structure among the MNPs and QCT [12], respectively.

### 3.2. XRD Analysis

The XRD patterns of QCT, MNPs, and QCT@MNPs are shown in Figure 1c. The characteristic crystalline peaks of QCT were observed at 10.78°, 12.44°, 16.12°, 23.84°, and 27.38°, which indicated the crystalline nature of QCT [27]. For MNPs, the characteristic peak appeared at 18.06°, which is attributed to the amorphous structure of MNPs [28]. However, when the MNPs were combined with QCT, the intensity of the peak decreased, and the crystal characteristics of QCT nearly disappeared, which indicated that the encapsulated QCT in Q@MNPs was in an amorphous form. It is possible that the original crystal structure of QCT was disrupted, and Π−Π stacking and hydrogen bonds were formed between the MNPs and QCT, which confirmed the results of the FT-IR analysis. A similar encapsulated QCT transformed into an amorphous form was reported in the literature. Zou et al. [7] encapsulated QCT in biopolymer-coated zein nanoparticles, and the diffraction pattern was similar to that of the plain nanoparticles, which showed that the encapsulated QCT was in an amorphous form. In addition, Gan et al. also found that after embedding plant sterols in sodium caseinate/pectin-based nanoparticles, plant sterols transitioned from an ordered crystalline state to an amorphous form [29]. 

### 3.3. TGA Analysis

The thermal stability of Q@MNPs was determined using TGA analysis (Figure 1d). The first stage of the weight loss of QCT, MNPs, and Q@MNPs occurred below 120 °C, which was attributed to the evaporation of water [30]. The second stage ranged from 120 to 370 °C, and a sharp decline in QCT appeared. This was attributed to pyrolysis, resulting in the polymer segmentation and degradation of the QCT [31]. From the DTG pattern of the QCT, there was an obvious peak at about 351 °C, showing a point of temperature where a large-scale weight loss occurred. According to the initial weight of the sample, the mass of QCT was lost about 34.36%, whereas those of MNPs and Q@MNPs were 23.12% and 22.15%, respectively. However, the thermal stability of Q@MNPs significantly improved (*p* < 0.05) when QCT was combined with MNPs. The final stage ranged from 370 to 600 °C, which was ascribed to the carbonization of the materials [32]. The weight of QCT, MNPs, and Q@MNPs finally lost about 53.55%, 44.51%, and 45.3%, respectively. Notably, the thermal decomposition patterns of the MNPs were not affected by QCT. However, the TGA curve of Q@MNPs was slightly lower than that of MNPs at the later period of carbonization, which can be explained by the thermal decomposition of QCT in the Q@MNPs, which caused slight weight loss [12]. In general, from the DTG curve, the overall curve trends of Q@MNPs were similar to those of MNPs. It can be concluded that the Q@MNPs improved the thermal stability of QCT.

### 3.4. TEM Analysis, Particle Size, Encapsulation Efficiency and Zeta Potential

The TEM analysis was used to evaluate the microscopic morphology of Q@MNPs. The size distribution of Q@MNPs was calculated based on the TEM observations. As can be observed in Figure 1e,f, the Q@MNPs exhibited a spherical structure and diameter with an average size of 26.68 ± 5.65 nm. The PDI of Q@MNPs was 0.259, which indicated the uniformity of Q@MNPs in a dispersion [33]. The Zeta potential and encapsulation efficiency of Q@MNPs were estimated. As observed in Figure 1g, the encapsulation efficiency of QCT loaded in Q@MNPs in this study was 43.78 ± 3.61%. It was reported that the stability of nanoparticle solution is dependent on the surface charge of the nanoparticle and higher zeta potential (absolute value) of electrostatic repulsion, and greater zeta potential manifested stronger electrostatic repulsion between nanoparticles [34]. The Zeta potentials of MNPs and Q@MNPs were −28.57 ± 0.46 and −35.87 ± 0.75 mV, respectively. As a result, the Zeta potential of Q@MNPs was greater than that of MNPs, which indicated that the encapsulation of QCT in MNPs could improve the spatial stability of Q@MNPs.

### 3.5. Photothermal Performance

The UV-vis absorption spectra of the QCT, MNPs, and Q@MNPs are shown in Figure 2a. This indicated that MNPs and Q@MNPs exhibited wide and strong absorption in the NIR region. After being irradiated (808 nm laser, 0.25 W/cm^2^) for 10 min (Figure 2b,c), the temperature of QCT, MNPs, and Q@MNPs increased to 64.4 °C, 121.6 °C, and 123.3 °C, respectively. MNPs can transform NIR light into heat and have high photothermal energy conversion efficiency in previous reports [8]. Moreover, there were no significant changes compared with MNPs and Q@MNPs, indicating that the photothermal performance was unaffected by the encapsulation of QCT. Next, in order to further evaluate the photothermal stability of Q@MNPs, heating and cooling cycles were performed four times (Figure 2d). After four cycles, the maximum temperature and change trend of Q@MNPs were not significantly changed (*p* < 0.05), which suggested that the Q@MNPs have good photothermal stability. Therefore, the Q@MNPs exhibited good photothermal properties and photothermal stability, which can be used as photothermal antibacterial nanomaterials in the food industry.

### 3.6. Antioxidant Activity of Q@MNPs

The DPPH and ABTS radical scavenging experiments were used to evaluate the antioxidant activities of MNPs and Q@MNPs. As can be seen in Figure 3a,b, when the concentration of MNPs increased from 0.05 to 0.8 mg/mL, the DPPH radical scavenging activities significantly improved (*p* < 0.05) from 4.85 ± 0.08 to 70.21 ± 0.74%, and the ABTS radical scavenging activities ranged from 21.89 ± 0.41 to 80.62 ± 0.43%. The antioxidant activity of MNPs is due to the intramolecular non-covalent electrons that can interact with free radicals [35]. As shown in Figure 3b, for Q@MNPs, with the concentration of Q@MNPs increased, the DPPH and ABTS radical scavenging ability significantly increased (*p* < 0.05) from 31.75 ± 0.11 to 86.98 ± 0.71%, 30.51 ± 0.84 to 88.26 ± 0.64%, respectively. Compared with MNPs, the DPPH and ABTS radical scavenging capacity of Q@MNPs at different concentrations was higher than that of MNPs. This can be ascribed to QCT promoting the antioxidant ability of Q@MNPs. The antioxidant ability of QCT was attributed to polyphenolic compounds, and QCT exhibited the ability to scavenge reactive oxygen species and form the equivalent hydrazine through the potential of hydrogen atoms [36]. According to Nalini et al. [37], encapsulating QCT into alginate/chitosan nanoparticles showed higher antioxidant activity than QCT alone. Hence, the Q@MNPs with high antioxidant activity can be used as food antioxidants in food processing.

### 3.7. Antibacterial Activity

The antibacterial activity of Q@MNPs against *E. coli* and *S. aureus* was next assessed in vitro using the method of plate counting. As can be seen in Figure 4a, after the treatment of each group with *S. aureus*, the survival rates of the QCT and QCT+NIR groups were 84.13%, and 84.42%, respectively, which may be attributed to the weak effects of QCT release [36]. The survival rates of the MNPs and MNPs+NIR groups were 96.08%, and 63.32%, respectively. This suggests that the MNPs may have some antibacterial activities against *S. aureus*. However, the antibacterial properties of *S. aureus* in the MNPs+NIR group significantly (*p* < 0.05) improved. This may be because MNPs are distributed in the cell walls and membrane of bacteria. The high temperature of MNPs produced from irradiation at 808 nm can later damage the bacteria, resulting in the release of the cytoplasmic matrix, thereby affecting its metabolism [38]. The survival rates of the Q@MNPs and Q@MNPs+NIR groups were 75.27% and 0.12%, respectively, suggesting that the photothermal effects of Q@MNPs combined with QCT resulted in good antibacterial activities against *S. aureus*. This could be attributed to the QCT binding to MNPs via Π−Π and hydrogen bonding interactions, and the photothermal effects produced high temperatures and weakened the interactions, resulting in a greater release of QCT [39]. Next, Figure 4b shows the bacteriostatic efficacy of each treatment group for *E. coli*. The survival rates of the QCT, QCT+NIR, MNPs, MNPs+NIR, Q@MNPs, and Q@MNPs+NIR groups were 90.61%, 87.77%, 82.76%, 68.86%, 68.29%, and 1.14%, respectively. Compared with *S. aureus*, MNPs showed stronger antibacterial activity against *E. coli*, which was similar to the report of Roy et al. [40]. Moreover, when Q@MNPs were combined with NIR, they showed excellent antibacterial effects on *E. coli*. Therefore, the excellent photothermal performance of Q@MNPs combined with QCT resulted in excellent antibacterial performance against *S. aureus* and *E. coli*. This can be used as a photothermal and antibacterial material for food processing.

### 3.8. In Vitro QCT Release

The QCT in vitro release behavior of Q@MNPs was investigated at pH 4.5 and 7.2 during 24 h. As illustrated in Figure 5a, the release of Q@MNPs exhibited three stages, including burst release, steady release, and release equilibrium. There was a fast release of QCT from Q@MNPs at pH 7.2 in the first 4 h, which reached 22.74%, and then increased to 26.28% at the steady stage (12 h), and eventually reached 27.33% at the equilibrium stage (24 h). The initial fast release can be explained by the QCT crystals located close to the surface of Q@MNPs, resulting in rapid dissolution [4]. It can be observed that the cumulative release of QCT from Q@MNPs at pH 4.5 was lower than at pH 7.2. This is because carboxyl groups in pH 7.2 carried negative charges due to deprotonation, resulting in higher repulsive forces between these groups on the polymer surface, which facilitated the release of QCT from the internal structure of Q@MNPs. In acidic media (pH 4.5), the electrostatic effect was minimized due to the protonation of carboxyl groups, which increased the ability to retain QCT [1]. These results showed that the release of QCT from Q@MNPs may be affected by different pH values [21]. After irradiation with an 808 nm laser, the cumulative release of QCT from Q@MNPs significantly increased (*p* < 0.05) in the first 1 h in PBS solutions of pH 4.5 and 7.2, and the cumulative release of 24 h significantly improved (*p* < 0.05), which can be attributed to the photothermal properties of Q@MNPs weakening the Π−Π and hydrogen bonding interactions between QCT and MNPs, promoting the release of QCT from Q@MNPs [39]. At 24 h, the cumulative release of Q@MNPs at pH 4.5 and 7.2 under irradiating treatments reached 16.31% and 35.36%, respectively. These results indicated that Q@MNPs exhibited fantastic NIR-responsive drug release behavior, which can be applied as photothermal bactericides in food preservation. Therefore, Q@MNPs with NIR/pH dual-responsive have great potential for drug delivery of QCT.

### 3.9. In Vitro Cytotoxicity Assay

The relative cell viability of L929 cells incubated with different concentrations of Q@MNPs using the CCK−8 assay is shown in Figure 5b. The relative cell viability of L929 cells was 86.16% and 74.04% at concentrations of 50 μg/mL and 100 μg/mL, respectively. In addition, when the concentration of Q@MNPs increased to 200 and 500 μg/mL, the relative cell viability decreased to 63.72% and 46.57%. According to previous research, the MNPs produced by synthetic methods showed good biocompatibility. Yuan et al. found that PDA60-PEG-NH_2_ nanoparticles exposed to L929 cells presented excellent relative cell viability exceeding 90% when the concentration was up to 1.0 mg/mL [41]. Ozlu et al. also stated that MNPs have no significant cytotoxicity to L929 cells [42]. Moreover, compared with the cell cytotoxicity of QCT, George et al. reported that a QCT concentration of 81 μg/mL results in 50% L929 cell death [2]. Therefore, the concentration of QCT increased with the concentration of Q@MNPs from 50 to 500 μg/mL, which may result in the reduced relative cell viability of L929 cells. These results indicated that the Q@MNPs presented good biocompatibility with L929 cells at concentrations below 100 μg/mL [43]. In conclusion, the Q@MNPs at low doses can be applied in the field of food processing.

## 4. Conclusions

In this work, Q@MNPs were successfully fabricated through the oxidative self-polymerization of dopamine in an alkaline aqueous solution combined with the absorption of QCT via Π−Π stacking and hydrogen bonding interactions. FT-IR, XRD, and Zeta potential analyses indicated that QCT was successfully loaded onto MNPs. The TGA analysis showed the enhanced thermal stability of QCT compounded with MNPs. The photothermal experiments revealed that Q@MNPs had excellent photothermal performance and stability, which could effectively sterilize *E*. *coli* and *S. aureus* after irradiation with an 808 nm NIR laser. Meanwhile, the encapsulation of QCT also improved the antioxidation performance of MNPs. Moreover, the Q@MNPs showed that the pH and NIR stimulated the sustained release of QCT. The relative cell viability of L929 cells results showed that the Q@MNPs at low doses (<100 μg/mL) presented good biocompatibility. To sum up, the as-prepared Q@MNPs exhibited improved antioxidant activity, photothermal antimicrobial activity, and NIR/pH dual-responsive drug release, which has great potential as a promising functional drug delivery platform in the food industry.

## Figures and Tables

**Figure 1 foods-12-04232-f001:**
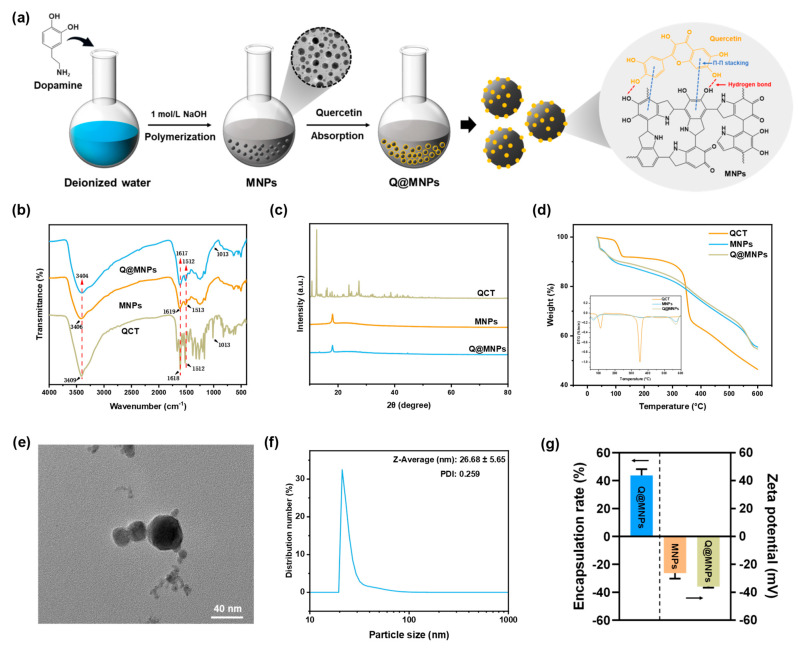
Schematic illustration of the procedure for the preparation of Q@MNPs (**a**); FT-IR spectra of QCT, MNPs, and Q@MNPs (**b**); XRD pattern of QCT, MNPs, and Q@MNPs (**c**); TGA curve of QCT, MNPs, and Q@MNPs (**d**); TEM image of Q@MNPs (scale bar: 40 nm) (**e**); particle size distribution of Q@MNPs (**f**); encapsulation rate and Zeta potential (**g**) of Q@MNPs.

**Figure 2 foods-12-04232-f002:**
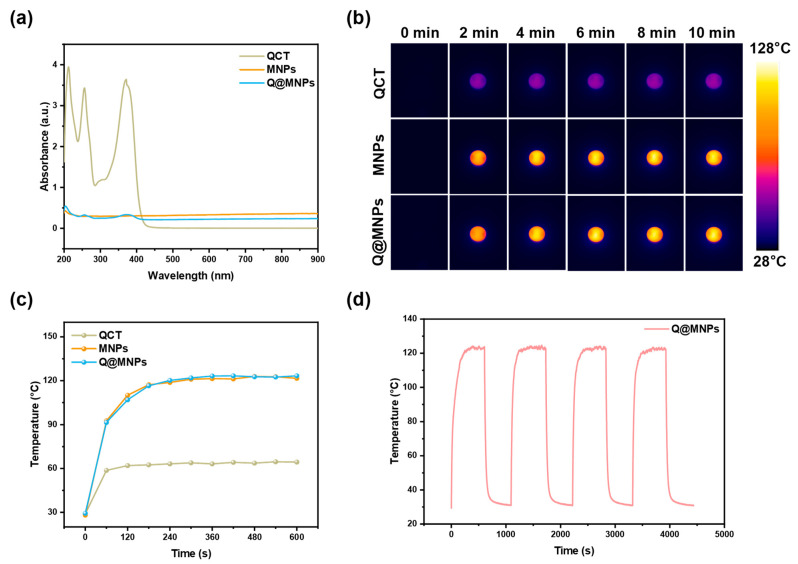
UV-vis absorbance (**a**); the photothermal image of QCT, MNPs, and Q@MNPs exposed to NIR laser at 808 nm (0.25 W/cm^2^) for 10 min (**b**); temperature changes in QCT, MNPs, and Q@MNPs irradiated using an 808 nm laser (**c**); and irradiated for four on/off cycles (**d**).

**Figure 3 foods-12-04232-f003:**
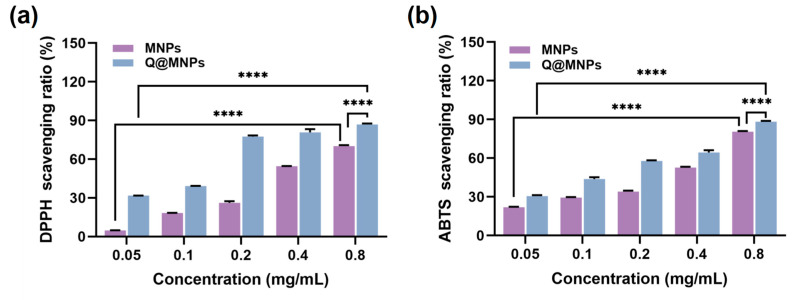
The DPPH radical scavenging ratio (**a**) and ABTS radical scavenging ratio of MNPs and Q@MNPs with increasing concentration from 0.05 to 0.8 mg/mL (**b**). Data presented as mean ± SD, **** *p* < 0.0001.

**Figure 4 foods-12-04232-f004:**
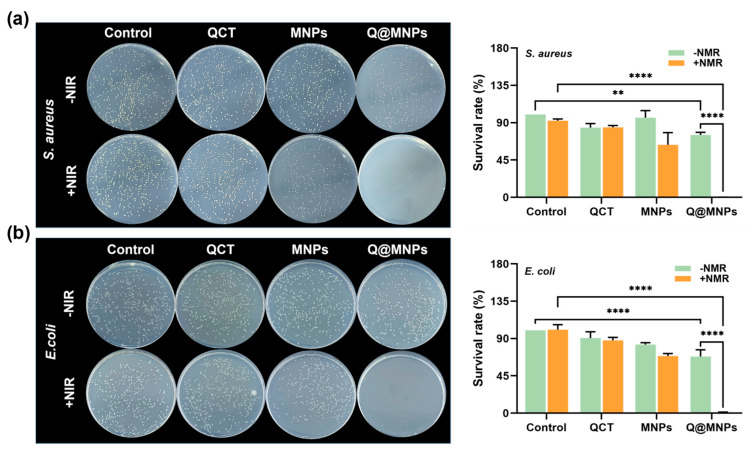
Photographs and survival rate (%) of *S. aureus* (**a**) and *E. coli* (**b**) treated with QCT, MNPs, and Q@MNPs. Data presented as mean ± SD, ** *p* < 0.01, **** *p* < 0.0001.

**Figure 5 foods-12-04232-f005:**
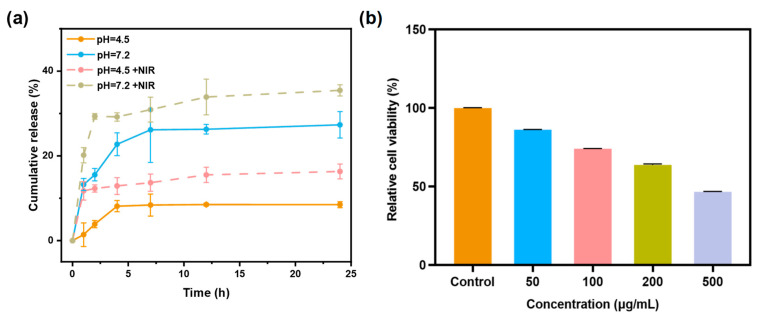
The cumulative release of QCT from Q@MNPs treated at pH = 4.5 (−NIR/+NIR) and pH = 7.2 (−NIR/+NIR) (**a**), and relative cell viability of L929 cells exposed to increasing concentrations of Q@MNPs from 50 to 500 μg/mL (**b**).

## Data Availability

The data used to support the findings of this study can be made available by the corresponding author upon request.

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
