# Peer review of "Biosynthesis of Quercetin-Loaded Melanin Nanoparticles for Improved Antioxidant Activity, Photothermal Antimicrobial, and NIR/pH Dual-Responsive Drug Release"

_foods, 2023, doi:10.3390/foods12234232_

Round 1

Reviewer 1 Report

Comments and Suggestions for Authors

This manuscript discuss about the Biosynthesis of quercetin-loaded melanin nanoparticles for improved antioxidant activity, photothermal antimicrobial and NIR/pH dual-responsive drug release. An interesting knowledge has been reported. however, the following comments should be addressed before acceptance

Comments

-the author should mention the unique significance of melanin NPs in the introduction part

- Authors have used quercetin as a synthesizing agent. however several papers also reported the use quercetin-loaded NPs. hence the authors must emphasize the novelty of the manuscript

-authors should mention about the method used to analysis FTIR such as kBr pellet method or ATR method

- . Zeta potential plays very important role, hence it is suggested to mention the clear experiment method used to analysis zeta potential sizer

-how does concentration effects the biological activity of MNPs

-biocompatibility of the mnps should be discussed in the manuscript

There are some typological errors are present in the that should be carefully revised in this manuscript

After addressing all the comments this manuscript can be acceptable for further progress

-

Comments on the Quality of English Language

 Minor editing of English language required

Author Response

Dear Reviewer

Thank you very much for your hard work in processing our manuscript (foods-2726380) and giving us an opportunity of the revise submission. We have added more results and discussions to address your comments in detail in the revised manuscript. After carefully studying the comments and your advice, we have made corresponding changes to the paper. All modifications are marked in red.

Responses to Reviewer #1

Comments 1: the author should mention the unique significance of melanin NPs in the introduction part.

Response: Thank you very much for your valuable comment. We have added the unique significance of melanin NPs in the introduction part, which can be found in Line 67-73, Page 2.

Comments 2: Authors have used quercetin as a synthesizing agent. however several papers also reported the use quercetin-loaded NPs. hence the authors must emphasize the novelty of the manuscript.

Response: Thank you very much for your valuable comment. Compared with previous reports, we have emphasized the novelty of the manuscript, which can be found in Line 81-86, Page 2.

Comments 3: authors should mention about the method used to analysis FTIR such as kBr pellet method or ATR method.

Response: Thank you very much for your valuable comment. We have mentioned the method used to analyze FTIR was KBr tableting method, which can be found in Line 138-139, Page 3.

Comments 4: Zeta potential plays very important role, hence it is suggested to mention the clear experiment method used to analysis zeta potential sizer.

Response: Thank you very much for your valuable comment. We have added the experiment method used to analyze Zeta potential sizer in detail, which can be found in Line 153-156, Page 4.

Comments 5: how does concentration effects the biological activity of MNPs.

Response: Thank you very much for your valuable comment. In this study, when the concentration increased from 0.05 to 0.8 mg/mL, the DPPH and ABTS radical scavenging ability significantly enhanced (p < 0.05) from 4.85 ± 0.08 to 70.21 ± 0.74%, 21.89 ± 0.41 to 80.62 ± 0.43%, respectively. The increased concentration of MNPs improved the abilities against DPPH and ABTS radical scavenging. The antioxidant activity of MNPs is due to the intramolecular non-covalent electrons that can interact with free radicals. Due to the MNPs almost having no antibacterial ability, the concentration affects little on the antibacterial properties of MNPs, but combined with NIR irradiation, the antibacterial properties against E. coli and S. aureus improved because of the photothermal performance of MNPs. In previous research, when the concentration of MNPs improved, the relative cell viability was reduced, but it maintained good biocompatibility and presented excellent relative cell viability exceeding 90% (https://doi.org/10.1021/acs.langmuir.2c01650).

Comments 6: biocompatibility of the mnps should be discussed in the manuscript.

Response: Thank you very much for your valuable comment. We have further discussed the biocompatibility of the MNPs in the discussion, which can be found in Line 426-440, Page 12.

Comments 7: There are some typological errors are present in the that should be carefully revised in this manuscript.

Response: Thank you very much for your valuable comment. We have checked the whole manuscript and some typological errors were revised, all modifications are marked in red.

Thank you again for all your suggestions. We hope that all these changes fulfill the requirements to make the manuscript acceptable for publication in foods and I am looking forward to hearing from you soon.

Thank you and best regards.

Sincerely,

Jie Pang

College of Food science

Fujian Agriculture and Forestry University

Shangxiadian Road 15#, Cangshan District, Fuzhou, 350002

E-mail: pang3721941@163.com

Reviewer 2 Report

Comments and Suggestions for Authors

Review report

1. I will suggest changing the short form “Q” used for quercetin to a pronounced notation such as “QCT”.

2. Line 13: Please remove the word “drug”.

3. The authors must provide PDI in the study of size distribution.

4. The subsection 2.2. is quite confusing, while preparing melanin nanoparticles (MNPs), the authors used dopamine, can the authors explain this?

5. Why the authors did not use HPLC for the in vitro release study of quercetin, as HPLC is more accurate and precise?

6. In the results and discussion section of FTIR, please provide some more details on the peak shifting after the attachment of Quercetin.

7. Labeling all the significant peaks in the FTIR spectra is suggested for better understanding.

8. There are numerous English language errors and grammatical mistakes.

Overall, the manuscript’s quality does not encourage me to recommend it for publication due to the number of flaws. The study design needs to be improved, the depth of work (particularly in the context of the biological assays) is not up to the mark, and the English language is not appropriate. I will recommend reconsideration of the manuscript after extensive revision.

Comments on the Quality of English Language

English editing is required.

Author Response

Dear Reviewer

Thank you very much for your hard work in processing our manuscript (foods-2726380) and giving us an opportunity of the revise submission. We have added more results and discussions to address your comments in detail in the revised manuscript. After carefully studying the comments and your advice, we have made corresponding changes to the paper. All modifications are marked in red.

Responses to Reviewer #2

Comments 1: I will suggest changing the short form “Q” used for quercetin to a pronounced notation such as “QCT”.

Response: Thank you very much for your valuable comment. We have changed the short form “Q”as“QCT”in the whole manuscript, all modifications are marked in red.

Comments 2: Line 13: Please remove the word “drug”.

Response: Thank you very much for your valuable comment. We have removed the word “drug”, which can be found in Line 17, Page 1.

Comments 3: The authors must provide PDI in the study of size distribution.

Response: Thank you very much for your valuable comment. We used a Litesizer 500 (Anton Parr, Germany) to measure the Particle size and polydispersity index (PDI) of Q@MNPs. The PDI of Q@MNPs size distribution was 0.259 in this study. We updated the latest version of Figure 1f and added the discussion of PDI, which can be found in Line315-316, Page 8.

Comments 4: The subsection 2.2. is quite confusing, while preparing melanin nanoparticles (MNPs), the authors used dopamine, can the authors explain this?

Response: Thank you very much for your valuable comment. In this study, we used green synthetic methods to prepare MNPs, the dopamine hydrochloride can oxidize in alkaline aqueous solution to form MNPs. Then the MNPs loaded with QCT to obtain Q@MNPs. We added the present of dopamine in the introduction, which can be found in Line 67-70, Page 2.

Comments 5: Why the authors did not use HPLC for the in vitro release study of quercetin, as HPLC is more accurate and precise?

Response: Thank you very much for your valuable comment. In this study, the method of UV−vis spectrophotometer was also used to evaluate the in vitro release study of quercetin. UV-vis spectrophotometry is usually used to measure the absorbance of drug solutions and determine the concentration of drugs. Specially, UV−vis spectrophotometer has been widely used for in vitro release study of quercetin. For examples, Zhang et al. prepared quercetin-loaded zein nanoparticles coated with dextrin-modified anionic polysaccharides, the release rate of quercetin was determined by UV−vis spectrophotometer (https://doi.org/10.1016/j.foodchem.2023.135736). Zhang et al. encapsulated quercetin into sodium caseinate decorating on shellac nanoparticles as a stabilizer, the simulated gastrointestinal digestion in vitro was evaluated using UV−vis spectrophotometer (https://doi.org/10.1016/j.foodchem.2022.133580). Ma et al. based on complex coacervation and fabricated co-delivery of curcumin and quercetin in the bilayer structure, the curcumin and quercetin content in the complexes at different release stage were determined using the method of UV−vis spectrophotometer (https://doi.org/10.1016/j.foodhyd.2023.108907).

Comments 6: In the results and discussion section of FTIR, please provide some more details on the peak shifting after the attachment of Quercetin.

Response: Thank you very much for your valuable comment. We have provided some more details on the peak shifting after the attachment of Quercetin, which can be found in Line 255-268, Page 6.

Comments 7: Labeling all the significant peaks in the FTIR spectra is suggested for better understanding.

Response: Thank you very much for your valuable comment. We have Labeled all the significant peaks in the FTIR spectra, which can be found in Figure 1b, Page 8.

Comments 8: There are numerous English language errors and grammatical mistakes.

Response: Thank you very much for your valuable comment. We have check the whole manuscript and English language errors and grammatical mistakes have been revised.

Thank you again for all your suggestions. We hope that all these changes fulfill the requirements to make the manuscript acceptable for publication in foods and I am looking forward to hearing from you soon.

Thank you and best regards.

Sincerely,

Jie Pang

College of Food science

Fujian Agriculture and Forestry University

Shangxiadian Road 15#, Cangshan District, Fuzhou, 350002

E-mail: pang3721941@163.com

Reviewer 3 Report

Comments and Suggestions for Authors

The current experiment has been well designed and implemented and has had promising results. Nevertheless, a similar work has been published before in the field of synthesis of quercetin-loaded melanin nanoparticles and have investigated its antibacterial effects, which unfortunately the authors have not mentioned this reference, so it is necessary to present the results of this article and other similar works at the end of the introduction and clearly state the novelty of their work in relation to these reference.

Comments on the Quality of English Language

Minor editing of English language required

Author Response

Dear Reviewer

Thank you very much for your hard work in processing our manuscript (foods-2726380) and giving us an opportunity of the revise submission. We have added more results and discussions to address your comments in detail in the revised manuscript. After carefully studying the comments and your advice, we have made corresponding changes to the paper. All modifications are marked in red.

Responses to Reviewer #3

Comments 1: The current experiment has been well designed and implemented and has had promising results. Nevertheless, a similar work has been published before in the field of synthesis of quercetin-loaded melanin nanoparticles and have investigated its antibacterial effects, which unfortunately the authors have not mentioned this reference, so it is necessary to present the results of this article and other similar works at the end of the introduction and clearly state the novelty of their work in relation to these reference.

Response: Thank you very much for your valuable comment. We have cited relevant literature and presented the results of this article and other similar works at the end of the introduction. Wang et al. extracted natural melanin to produce synthesis of polydopamine nanoparticles and modified the surface of quercetin-melanin, which exhibited excellent antibacterial and anti-tumor properties (https://doi.org/10.1002/jbm.b.34813). Their work improved the bioavailability of QCT and expanded the application of natural MNPs. The modifications can be found in Line 81-86, Page 2.

Thank you again for all your suggestions. We hope that all these changes fulfill the requirements to make the manuscript acceptable for publication in foods and I am looking forward to hearing from you soon.

Thank you and best regards.

Sincerely,

Jie Pang

College of Food science

Fujian Agriculture and Forestry University

Shangxiadian Road 15#, Cangshan District, Fuzhou, 350002

E-mail: pang3721941@163.com

Reviewer 4 Report

Comments and Suggestions for Authors

This manuscript discusses the biological effects of the quercin complex loaded in melanin nanoparticles as a potential process used in food processing and as an in vivo drug delivery system.

Many analyses (FTIR, XRD, TGA, TEM, ZETA potential...) were initiated to characterize this complex (Q@MNPs). The stability and antioxidant, antibacterial, and biocompatibility activities were also evaluated.

Some publications discuss this topic of the evaluated manuscript, Quercetin being loaded in various types of nanoparticles (chitosan, polymers...), including melanin (which the authors capture very well in the introduction), but very limited in terms of its use in amorphous form.

The authors discuss the Q@MNPs complex as an in vivo drug delivery system, why is there no evidence for this?

 In vivo experiments can offer big surprises regarding the effectiveness of this complex compared to in vitro studies.

Why did the authors choose to test the effectiveness of the complex on S. aureus?

This germ is not involved in inducing foodborne infections, it could have been chosen next to E. coli, Salmonella, Campylobacter, Cryptosporidium, Cyclospora spp, Giardia, and Listeria monocytogenes.

On the other hand, in the studies discussing complexed quercetin, the research is done only on E. coli.

Why weren't other foodborne pathogen germs chosen from those presented to strengthen the originality of the article?

I agree with the publication of the manuscript after the authors answer the questions.

Reviewer 5 Report

Comments and Suggestions for Authors

The article ‘Biosynthesis of quercetin-loaded melanin anoparticles for improved antioxidant activity, photothermal antimicrobial and NIR/pH dual-responsive drug release’, presented for review is interesting. However, some issues require clarification. Below are some comments that may improve the article.

- Reliability and repeatability of measurements require that the standard deviation is not greater than 5%. Meanwhile, with the antioxidant activity measured against the DPPH radical for a result of 9.86, the authors provide a standard deviation of 2.52, which is over 25%. How to explain such a large standard deviation. (Line 318-321: As can be seen in Figure 3a-b, when the concentration of MNPs increased from 0.05 to 0.8 mg/mL, the DPPH radical scavenging activities significantly improved (p < 0.05) from 9.86 ± 2.52 to 72.40 ± 2.77 %, and the ABTS radical scavenging activities were from 22.54 ± 1.1 to 82.35 ± 2.63%.)

- There may also be some doubts regarding the values of standard deviations provided later in the discussion of antioxidant activity. (Line-323-325) As shown in Fig. 3b, for Q@MNPs, with the concentration of Q@MNPs increased, the DPPH and ABTS radical scavenging ability significantly enhanced (p < 0.05) from 30.48 ± 1.9 to 87.01 ± 0.75 %, 32.14 ± 2.19 to 88.08 ± 0.72%, respectively.

Author Response

Dear Reviewer

Thank you very much for your hard work in processing our manuscript (foods-2726380) and giving us an opportunity of the revise submission. We have repeated some experiments to address your comments in detail in the revised manuscript. After carefully studying the comments and your advice, we have made corresponding changes to the paper. All modifications are marked in red.

Responses to Reviewer #4

Comments 1: Reliability and repeatability of measurements require that the standard deviation is not greater than 5%. Meanwhile, with the antioxidant activity measured against the DPPH radical for a result of 9.86, the authors provide a standard deviation of 2.52, which is over 25%. How to explain such a large standard deviation. (Line 318-321: As can be seen in Figure 3a-b, when the concentration of MNPs increased from 0.05 to 0.8 mg/mL, the DPPH radical scavenging activities significantly improved (p < 0.05) from 9.86 ± 2.52 to 72.40 ± 2.77 %, and the ABTS radical scavenging activities were from 22.54 ± 1.1 to 82.35 ± 2.63%.)

Response: Thank you very much for your valuable comment. We have repeated the measurements of the antioxidant activity of MNPs against DPPH and ABTS radical scavenging. The results were analyzed again and the figures were updated with the latest version (Figure 3). The analysis of results can be found in Line 349-350, Page 9.

Comments 2: There may also be some doubts regarding the values of standard deviations provided later in the discussion of antioxidant activity. (Line-323-325) As shown in Fig. 3b, for Q@MNPs, with the concentration of Q@MNPs increased, the DPPH and ABTS radical scavenging ability significantly enhanced (p < 0.05) from 30.48 ± 1.9 to 87.01 ± 0.75 %, 32.14 ± 2.19 to 88.08 ± 0.72%, respectively.

Response: Thank you very much for your valuable comment. We have repeated the measurements of the antioxidant activity of Q@MNPs against DPPH and ABTS radical scavenging. The results were analyzed again and the figures were updated with the latest version (Figure 3). The analysis of results can be found in Line 354, Page 9. 

Thank you again for all your suggestions. We hope that all these changes fulfill the requirements to make the manuscript acceptable for publication in foods and I am looking forward to hearing from you soon.

Thank you and best regards.

Sincerely,

Jie Pang

College of Food science

Fujian Agriculture and Forestry University

Shangxiadian Road 15#, Cangshan District, Fuzhou, 350002

E-mail: pang3721941@163.com
